# Solvent-Dependent Structures of Natural Products Based on the Combined Use of DFT Calculations and ^1^H-NMR Chemical Shifts

**DOI:** 10.3390/molecules24122290

**Published:** 2019-06-20

**Authors:** Saima H. Mari, Panayiotis C. Varras, Iqbal M. Choudhary, Michael G. Siskos, Ioannis P. Gerothanassis

**Affiliations:** 1H.E.J. Research Institute of chemistry, International Center for Chemical and Biological Sciences, University of Karachi, Karachi 7527, Pakistan; Saimahassanmari123@gmail (S.H.M.); iqbal.choudhary@iccs.edu (I.M.C.); 2Section of Organic Chemistry & Biochemistry, Department of Chemistry, University of Ioannina, GR-45110 Ioannina, Greece; panostch@gmail.com (P.C.V.); msiskos@uoi.gr (M.G.S.); 3Dr. Panjwani Center for Molecular Medicine and Drug Research, International Center for Chemical and Biological Sciences, University of Karachi, Karachi 7527, Pakistan

**Keywords:** chemical shifts, hydrogen bonding, DFT, natural products, NMR, X-ray diffraction

## Abstract

Detailed solvent and temperature effects on the experimental ^1^H-NMR chemical shifts of the natural products chrysophanol (**1**), emodin (**2**), and physcion (**3**) are reported for the investigation of hydrogen bonding, solvation and conformation effects in solution. Very small chemical shift of │Δδ│ < 0.3 ppm and temperature coefficients │Δδ/ΔΤ│ ≤ 2.1 ppb/K were observed in DMSO-d_6_, acetone-d_6_ and CDCl_3_ for the C(1)–OH and C(8)–OH groups which demonstrate that they are involved in a strong intramolecular hydrogen bond. On the contrary, large chemical shift differences of 5.23 ppm at 298 K and Δδ/ΔΤ values in the range of −5.3 to −19.1 ppb/K between DMSO-d_6_ and CDCl_3_ were observed for the C(3)–OH group which demonstrate that the solvation state of the hydroxyl proton is a key factor in determining the value of the chemical shift. DFT calculated ^1^H-NMR chemical shifts, using various functionals and basis sets, the conductor-like polarizable continuum model, and discrete solute-solvent hydrogen bond interactions, were found to be in very good agreement with the experimental ^1^H-NMR chemical shifts even with computationally less demanding level of theory. The ^1^H-NMR chemical shifts of the OH groups which participate in intramolecular hydrogen bond are dependent on the conformational state of substituents and, thus, can be used as molecular sensors in conformational analysis. When the X-ray structures of chrysophanol (**1**), emodin (**2**), and physcion (**3**) were used as input geometries, the DFT-calculated ^1^H-NMR chemical shifts were shown to strongly deviate from the experimental chemical shifts and no functional dependence could be obtained. Comparison of the most important intramolecular data of the DFT calculated and the X-ray structures demonstrate significant differences for distances involving hydrogen atoms, most notably the intramolecular hydrogen bond O–H and C–H bond lengths which deviate by 0.152 tο 0.132 Å and 0.133 to 0.100 Å, respectively, in the two structural methods. Further differences were observed in the conformation of –OH, –CH_3_, and –OCH_3_ substituents.

## 1. Introduction

The single-crystal X-ray diffraction has been the most widely used method for structural analysis in the solid state [1,2,3]. In macromolecular crystallography, it may be argued that the structural data bears some resemblance to the solution structure, since the crystals frequently contain significant numbers of solvent molecules, usually water [4,5]. In organic chemistry crystallography, however, the incorporation of solvent molecules can be, in most cases, very limited. Extrapolation, therefore, of the molecular conformation in the crystal to possible conformations in solution, is very problematic since crystal packing interactions can stabilize conformers that are rarely encountered in solution.

NMR spectroscopy is among the primary methods for investigating structure and dynamics of complex molecules in solution [6]. Several NMR parameters such as chemical shifts [7,8,9], temperature and solvent effects on chemical shifts [10,11], the NOE phenomenon [12,13], and spin-spin couplings [6,14,15] have been utilized in order to establish empirical correlation with structural data. The development of quantum chemical methods for calculating NMR parameters, with emphasis on chemical shifts, as well as advances in computational techniques and computer power, have led to an increasing number of studies that combine calculation with experiment [16,17,18,19,20]. Several examples of using computed NMR chemical shifts to confirm proposed structures or to aid the reassignment of structures can be found in the literature [20,21,22,23,24,25]. To date, however, only a handful examples of organic molecules whose structures have been determined by both computation of NMR chemical shifts in solution and X-ray were reported [26,27,28,29,30,31]. Furthermore, to the best of our knowledge, no solvent-dependent DFT structures have so far been reported, thus, there is no sufficient experimental basis for assessing the accuracy of NMR solution structures.

Herein, a comparison of the solvent-dependent structures, based on the combined use of DFT calculations and ^1^H-NMR chemical shifts, and the single-crystal X-ray structures of three natural products, chrysophanol (**1**), emodin (**2**), and physcion (**3**) (Scheme 1), are reported in order to quantify the degree of similarity/ difference between the results obtained with these two structural methods and to ascertain the molecular and electronic origin of the differences. The three molecules selected belong to the anthraquinone family. Chrysophanol (**1**), also known as chrysophanic acid, was identified as a fungal metabolite which blocks the proliferation of colon cancer cells in vitro [32]. Emodin (**2**), derived from *Rheum emodi* (Himalyan rhubarb), possess a wide range of anticancer, anti-oxidant, hepatoprotective, anti-inflammatory and anti-microbial properties [33,34]. Physcion (**3**), also known as parietin, is a predominant cortical pigment of lichens in the genus *Caloploca*. Physcion (**3**) from marine-derived fungus *Microsporum* sp. was reported to induce apoptosis in human cervical carcinoma HeLa cells [35].

## 2. Results and Discussion

### 2.1. Experimental ^1^H-NMR Chemical Shifts

#### 2.1.1. Assignment of the Resonances

For the investigation of solvent and temperature effects on the ^1^H-NMR resonances, the unequivocal assignment of the resonances of the molecules under study should be achieved. The 1D ^1^H-NMR spectra of chrysophanol (**1**), emodin (**2**), and physcion (**3**) show extremely deshielded and sharp peaks which are attributed to the C(1)–OH and C(8)–OH protons (Table 1) which participate in a strong intramolecular hydrogen bond with the O=C(9) carbonyl oxygen atom. The planar configuration of the two six membered rings results in a significant deshielding [8,36,37,38,39,40,41,42] and reduction of the OH proton exchange rates [36,37,38,39,40,41,42]. This allowed the application of the ^1^H-^13^C HMBC NMR experiments to reveal long-range ^n^J(^13^C,^1^H) connectivities (Appendix A), thus, providing an unequivocal assignment of the C(1)–OH and C(8)–OH groups. The assignments of the ^1^H-NMR chemical shifts were, furthermore, confirmed by 1D-NOE and 1D-TOCSY experiments.

#### 2.1.2. Solvent Effects on ^1^H-NMR Chemical Shifts

Three solvents, CDCl_3_ (ε = 4.81), acetone-d_6_ (ε = 20.7) and DMSO-d_6_ (ε = 46.7) with significantly different dielectric constants, solvation and hydrogen bonding ability were selected to investigate the effect of solvents on intra- and inter- molecular hydrogen bond and conformation of the three natural products. The experimental data are reported in Table 1 using the same experimental conditions i.e., dilute solutions with concentration ≤2 mM at 298 K. The results clearly demonstrate that the C(1)–OH and C(8)–OH groups are involved in strong intramolecular hydrogen bond which is not affected by the solvent even in the case of DMSO, which is a solvent of high dielectric constant and solvation ability. Due to strong intramolecular hydrogen bond, the solvent molecules around the C(1)–OH and C(8)–OH protons are excluded, leading to a significantly reduced solvation and, thus, very small chemical shift solvent-dependence.

The C(3)–OH resonance of emodin (**2**) was found to be extremely broad and could hardly be distinguished from the baseline, contrary to the case of the C(1)-OH and C(8)-OH groups which participate in a strong intramolecular hydrogen bond. Addition of 2 μL of trifluoroacetic acid (TFA) resulted in very sharp resonances which allowed the accurate estimation of chemical shifts in DMSO-d_6_ and acetone-d_6_ (Figure 1)_._ The resulting chemical shifts in DMSO-d_6_ (δ = 11.41), acetone-d_6_ (δ = 10.21) and CDCl_3_ (δ = 6.18) clearly indicate that hydrogen bond between the C(3)–OH group and DMSO-d_6_ is more efficient than in acetone-d_6_ and significantly stronger than in CDCl_3_. This demonstrates the great sensitivity of ^1^H-NMR chemical shifts to both intra- and intermolecular hydrogen bond [26,27,28,29,30,40,41,42,43,44,45,46,47,48].

Surprisingly, a deuterium isotopic effect was observed, when the spectra were recorded in acetone-d_6_. Figure 2a shows a splitting of the peaks of the C(1)–OH and C(8)–OH groups which is attributed to the high content of residual D_2_O in the acetone-d_6_ solvent. In cases that the OH(*^2^*H) exchange rate is slow in the NMR time scale, two separate resonances are observed due to the proton and deuterium species, with relative intensities which depend on the H/^2^H fractionation ratio [36,49]. Several long range ^n^ΔH(^2^H) isotope effects have been reported in the literature and their magnitude is larger in cases of strong intramolecular hydrogen bond [43,50]. Figure 2b shows that, after the addition of 1–2 μL of H_2_O, the more deshielded resonances of the doublets were reduced significantly in intensity because of the substitution of ^2^H with the ^1^H isotope. This furthermore revealed that the long-range deuterium isotopic effect, ΔΗ(^2^H) = δ_Η_-δ_H/_^2^_H_, which is transmitted through the intramolecular hydrogen bond (Figure 2), is negative and the magnitude was found to be: chrysophanol C(1)–OH(^2^H) ≈ −14.0 ppb and C(8)–OH(^2^H) ≈ −14.0 ppb; emodin C(1)–OH(^2^H) ≈ −18.5 ppb and C(8)–OH(^2^H) ≈ −19.6 ppb and physcion C(1)–OH(^2^H) ≈ −19.4 ppb and C(8)–OH(^2^H) ≈ −18.0 ppb, where ppb denotes parts-per-billion, 10^−9^.

#### 2.1.3. Temperature-Dependence of ^1^H-NMR Chemical Shifts

The temperature coefficient of the chemical shifts of OH groups, Δδ/ΔT, can provide useful information to distinguish intra- and inter- molecular hydrogen bond which is complimentary to that obtained with the use of solvent effects [10,11,40,41,42]. Appendix A shows variation in the chemical shift of C(1)–OH, C(8)–OH and C(3)–OH of emodin (**2**) in CDCl_3_, acetone-d_6_ and DMSO-d_6_ at different temperatures. The OH resonances are shielded linearly as the temperature increases and this results in negative temperature coefficients with the coefficient of determination R^2^ > 0.997. In all cases, the Δδ/ΔT values of C(1)–OH and C(8)–OH were found to be very small and in the range of −0.4 to −2.1 ppb/K (Table 2) which demonstrate that these groups are involved in a strong intramolecular hydrogen bond with the C(9)=O carbonyl oxygen. These results are in excellent agreement with the very small variation of chemical shifts in various solvents which were found to be │Δδ│< 0.15 ppm (Table 1). Δδ/ΔT values of the C(3)–OH group of emodin (**2**) are in the range of −5.3 to −19.1 ppb/K (Table 2). Surprisingly, the Δδ/ΔT value in CDCl_3_ (−19.1 ppb/K) is significantly larger than those in DMSO-d_6_ and acetone-d_6_, although CDCl_3_ is a weak hydrogen bond donor solvent. This indicates that the C(3)–OH group is involved in strong solute-to-solute intermolecular hydrogen bond interaction although the solution concentration was <2 mM. This is in agreement with literature data on large temperature coefficients of NH chemical shifts in CDCl_3_ when intermolecular self-association is significant [51]. Interestingly, the line width of the C(3)–OH resonance in CDCl_3_ decreases by increasing the temperature, contrary to the case in acetone-d_6_ and DMSO-d_6_ (Appendix A). This implies that the mechanism of proton transfer in CDCl_3_ involves intermolecular solute-to-solute hydrogen bond which is broken by increasing the temperature. On the contrary, in acetone-d_6_ and DMSO-d_6_ the proton transfer involves the phenol-OH group and traces of water in the organic solution and, thus, the exchange rate increases by increasing the temperature.

### 2.2. Quantum Chemical Calculations

#### 2.2.1. DFT-Calculated vs. Experimental ^1^H-NMR Chemical Shifts in Solution with CPCM: Effects of Various Functionals and Basis Sets

Appendix A illustrates DFT-calculated [52] ^1^H NMR chemical shifts (δ_calc_) of chrysophanol (**1**), emodin (**2**), and physcion (**3**), with optimization of the structures at the B3LYP/6-31+G(d), ωB97XD/6-31+G(d), APFD/6-31+G(d), M06-2X/Def2TZVP, and TPSSh/TZVP level of theory, with CPCM [53] in CHCl_3_, acetone, and DMSO. The presence of various conformers of the C(3)–OH and C(3)–OCH_3_ substituents in emodin (**2**) and physcion (**3**) complicates the interpretation of the computed chemical shifts (see discussion below), therefore, chrysophanol (**1**) was selected to investigate the effect of the basis set on the optimized geometries and the quality of the ^1^H-NMR chemical shift calculations.

Figure 3 illustrates calculated ^1^H-NMR chemical shifts (at the GIAO/B3LYP/6-311+G(2d,p) level with CPCM in CHCl_3_) vs. experimental ^1^H-NMR chemical shifts of chrysophanol (**1**), with optimization of the structures using the above five functionals and the three basis sets. Very good linear regression correlation coefficients and standard deviations of the ^1^H-NMR chemical shifts were obtained for the various functionals and basis sets used (Table 3). Interestingly, calculations at the B3LYP/6-31+G(d) level performed better in terms of mean square error and slope than that at the TPSSh/TZVP level. It can, therefore, be concluded that accurate results may not require computationally demanding basis sets in order to obtain optimized geometries. This is in excellent agreement with the conclusion of a recent comprehensive review article that increasing basis set and computational time does not necessarily result in more accurate chemical shifts at least with B3LYP [20].

An intramolecular hydrogen bond of the C(1)–OH and C(8)–OH groups with the carbonyl CO oxygen was observed in the DFT calculated structures of chrysophanol (**1**), emodin (**2**), and physcion (**3**) in the three solvents studied. The computed ^1^H-NMR chemical shifts of the C(1)–OH and C(8)–OH groups were found in the range of δ11-12 in CHCl_3_, acetone, and in DMSO and in very good agreement with the experimental ^1^H-NMR chemical shifts (Appendix A). This demonstrates that the intramolecular hydrogen bond is independent of the solvents used. On the contrary, the solvent-dependent experimental ^1^H-NMR chemical shifts of the C(3)–OH group of emodin (**2**) strongly deviate from the computational data: δ_exp,acetone_ − δ_calc,acetone_ = 4.82 ppm and δ_exp,DMSO_ − δ_calc,DMSO_ = 6.01 ppm. This indicates that the continuum model is not appropriate for solvent exposed OH groups and, thus, it is necessary the incorporation of discrete molecules of solvent (discussed in Section 2.2.3)**_._**

#### 2.2.2. Effect of Conformation of Substituents on the Calculated ^1^H-NMR Chemical Shifts

The effect of the C(3)–OH and C(3)–OCH_3_ substituents of emodin (**2**) and physcion (**3**), on the calculated ^1^H-NMR chemical shifts, has been investigated in detail. Figure 4 illustrates the electronic energy (Hartree units) of emodin (**2**) as a function of the torsion angle φ = C(4)–C(3)–O(3)–H(3), at the B3LYP/6-31 + G(d) level (gas phase). As expected, a maximum of the electronic energy for a torsion angle φ = 90° was observed. The Gibbs energy of conformer A (φ = 0°) was found to be lower by 0.24 kcal/mol with respect to that with φ = 180°. Similar results were obtained with the various functionals and basis sets of Appendix A. This implies that there is an equilibrium of the two fast-interconverting, on the NMR time scale, conformers with nearly equal populations. Although the energy difference of the two conformers of emodin (**2**) is negligible (ΔG = 0.24 to 0.11 kcal/mol), their dipole moments differ significantly, due to the different orientation of the hydroxyl group (*μ_A_* = 2.8 D and *μ_Β_* = 0.4 D) (Appendix A).

Figure 5 illustrates the dependence of calculated GIAO ^1^H-NMR chemical shifts of emodin (**2**) as a function of torsion angle φ. A broad minimum of the computed δ(^1^Η) NMR chemical shifts of the OH group was observed for φ = 90° with a maximum shielding of ~1.8 ppm than in the in-plane conformers. The H(2) and H(4) protons, which are in ortho position with respect to the OH group, demonstrate a Karplus-like variation with a deshielding of ~0.4 ppm for φ = 90°. The chemical shift difference of the H(2) and H(4) protons for φ = 0° is 0.5 ppm and increases up to 0.95 ppm for φ = 180°. Of particular interest is the dependence of calculated chemical shifts of the C(1)–OH and C(8)–OH groups as a function of the φ angle. For φ = 0°, the C(1)–OH proton is more shielded than the C(8)–OH while for φ = 180° is more deshielded with a significant increase in the chemical shift difference of C(1)–OH and C(8)–OH, which can provide a criterion of the conformation of the OH substituent.

Figure 6 shows one of the molecular orbitals for conformer A of emodin (**2**). It can be seen clearly that there exists a bonding lobe embracing atoms H(3), O(3), C(3), C(2), C(1), O(1), and the hydrogen atom H(1) which participates in the intramolecular hydrogen bond. This affects the electron density at the hydrogen atom and, consequently, the corresponding chemical shift. Molecular orbital having similar bonding characteristics could not be found for conformer B. The intramolecular hydrogen bond distances for conformer A are almost identical (O(1)–H(1)∙∙∙O(9) = 1.703 Å and O(8)–H(8)∙∙∙O(9) = 1.704 Å), while for conformer B there is a significant difference (O(1)–H(1)∙∙∙O(9) = 1.695 Å and O(8)–H(8)∙∙∙O(9) = 1.704 Å). The shortest hydrogen bond distance of H(1) in conformer B is also reflected in the deshielding with respect to H(8) which can be interpreted taking into consideration the well-established linear relationship of δ(^1^H) vs. (O)H∙∙∙O distances [8,9,26,29].

Figure 7 illustrates the electronic energy (Hartree unit) of physcion (**3**) as a function of the torsion angle φ = C(4)–C(3)–O(3)–C(3′) of the OCH_3_ group. Contrary to the case of emodin, the Gibbs energy of conformer A (φ = 0°) was found to be higher by ~0.07 kcal/mol than that with φ = 180°. The maximum deshielding of the C(4)–H and C(2)–H occurs at φ = 60° and 120°, respectively. Of particular interest is the complex dependence of calculated chemical shifts of the C(1)–OH and C(8)–OH protons as a function of torsion angle φ (Appendix A). In the case of φ < 30°, the C(1)–OH proton is more deshielded than the C(8)–OH, and for φ > 30° becomes more deshielded with a maximum chemical shift difference of ~0.45 ppm for φ = 180°. This difference, therefore, can be used as a criterion of the conformation of the OCH_3_ substituent.

Figure 8 illustrates molecular orbital analysis of conformers A and B of physcion (**3**). Although the energy difference between the two conformers is negligible (ΔΕ = 0.23 kcal/mol, ΔG = 0.07 kcal/mol), the rotation of the methoxy group (–OCH_3_) affects differently the electron density at the hydrogen atom which is involved in the intramolecular hydrogen bond. The effect for conformer B is transmitted through a sigma (σ) bonding molecular orbital (MO) embracing all the atoms O(3), C(3), C(2), C(1), O(1), H(1), and O(9), while for conformer A an antibonding region between O(3) and C(3) is evident.

Due to rotation of the methyl group substituent in chrysophanol (**1**), emodin (**2**), and physcion (**3**), the minimum Gibbs energy conformational state is the one in which the C–H group is in the eclipsed conformation with respect to the planar aromatic system (Appendix A).

#### 2.2.3. Effect of Discrete Solvent Molecules on Intra- and Intermolecular Hydrogen Bond Interactions and Conformation of Substituents

As pointed out in Section 2.2.1, the continuum model is not appropriate for solvent-exposed OH groups, therefore, the effect of discrete solvation molecules was investigated in detail. For emodin (**2**) with one molecule of chloroform, optimized in continuum (CPCM-CHCl_3_), two conformers (A and B, Appendix A) which constitute minima on the potential energy surface (PES), were obtained with their energy difference being only 0.04 kcal/mol (Appendix A). A typical hydrogen bond was observed between the lone electron pair of the phenolic oxygen and the hydrogen of the chloroform with H∙∙∙O distances of 2.223 Å and 2.227 Å for conformers A and B, respectively. These values can be compared with a distance of 2.175 Å obtained for a 1:1 PhOH + CHCl_3_ complex [41]. Molecular orbital analysis for the two conformers A and B (Appendix A) shows that the rotation of the C(3)–OH hydroxyl group, which forms an intermolecular hydrogen bond with the chloroform molecule, will influence the chemical shift of proton C(1)–OH. For conformer A, this effect will be greater than the corresponding conformer B, since in the former case a bonding lobe embracing atoms, H(3), O(3), C(3), C(2), C(1), O(1), H(1), O(9), and C(9) is clearly seen, while for conformer B the corresponding lobe has an antibonding character. Consequently, the electron density at H(4) will be different in the two conformers and, hence, their chemical shifts.

The optimized-structures of conformers A and B of emodin (**2**) in continuum (CPCM-DMSO) with a single molecule of DMSO at the C(3)–OH group are shown in Figure 9. Conformer A exists in two local minima A_1_ and A_2_ with an energy difference of only 0.08 kcal/mol (ΔG ~ 0.04 kcal/mol). Both conformers A_1_ and A_2_ are more stable than conformer B (Appendix A). The stabilization of all conformers may be attributed to the strong intermolecular hydrogen bonds O–H∙∙∙O(S) of 1.639 Å (conformer A_1_), 1.653 Å (conformer A_2_) and 1.650 Å (conformer B). The C(3)–O–H∙∙∙O torsion angles were found to be 83.52° (conformer A_1_), −0.22° (conformer A_2_) and 20.04° (conformer B), while the two methyl groups of DMSO are nearly perpendicular and symmetrical to the planar aromatic system in conformers A_1_ and B. The hydrogen bond angles O–H∙∙∙O(S) were found to be 179.26° (conformer A_1_), 177.67° (conformer A_2_) and 177.29° (conformer B) and, thus, deviate slightly from linearity. The torsion angles O–H∙∙∙O=S were found to be 125.01° (conformer A_1_), 169.37° (conformer A_2_) and 178.14° (conformer B). Molecular orbital analysis demonstrates the existence of a bonding lobe embracing atoms O(3), C(3), C(2), C(1), O(1), H(1), O(9), and C(9) in conformer A_2_, while for conformers A_1_ and B the corresponding lobe has an antibonding character (Appendix A).

The optimized structures of conformers A and B of emodin (**2**) with a single molecule of acetone at the C(3)–OH group are shown in Appendix A. The stabilization of both conformers may be attributed to a relatively strong intermolecular hydrogen bond O–H∙∙∙O(C) of 1.737 Å and 1.735 Å for conformers A and B, respectively. The hydrogen bond angles O–H∙∙∙O(C) were found to be 175.26° (conformer A) and 175.88° (conformer B) and, thus, deviate slightly from linearity. The torsion angles O–H∙∙∙O=C were found to be 110.46° (conformer A) and 113.80° (conformer B) and the torsion angles C–O–H∙∙∙O(C) 139.09° (conformer A) and −134.7° (conformer B).

Appendix A illustrates calculated (δ_calc_, ppm) and δ_exp_ – δ_calc_
^1^H-NMR chemical shifts of emodin (**2**) complexes with a single molecule of DMSO, acetone, and CHCl_3_. The incorporation of discrete solvent molecule of DMSO and acetone induces a very significant variation in the chemical shift, due to intermolecular hydrogen bond of solvent molecule with the C(3)–OH group of emodin (**2**). This results in excellent agreement of the computed, at the GIAO_B3LYP/6-311+G(2d,p) (CPCM-DMSO) level, with the experimental chemical shifts with R^2^ = 0.9991 and 0.9992, slopes of 1.0347 and 1.0305, and mean square errors of 0.0909 and 0.0882 ppm, for conformers A_1_ and B, respectively (Figure 10a). Figure 10b illustrates calculated vs. experimental ^1^H-NMR chemical shifts of conformers A and B of emodin (**2**) in CPCM-DMSO. The results of the linear regression analysis (R^2^ = 0.7181 and 0.7822, slopes of 0.8309 and 0.8384 and mean square errors of 1.5026 and 1.5057 ppm for A and B conformers, respectively), clearly show a poor correlation mainly due to differences of the C(3)–OH chemical shifts.

A discrete molecule of DMSO was also placed in the vicinity of either the C(1)–OH or C(8)–OH groups (Appendix A). Very minor changes in the chemical shifts of protons of C(1)–OH and C(8)–OH groups were observed because C(1)–OH and C(8)–OH are involved in a strong intramolecular hydrogen bond with the oxygen atom of the O=C(9) group. The distance –(O)H∙∙∙O(S) of the optimized structure was found to be 2.699 Å which indicates that the discrete molecule of DMSO was pushed away from the C(1)–OH and C(8)–OH groups (Appendix A). This is in excellent agreement with the experimental NMR data that the formation of intramolecular hydrogen bond is solvent independent. Similar computational results were obtained when two molecules of DMSO were placed symmetrically in the vicinity of the intramolecular hydrogen bond.

#### 2.2.4. Comparison Between DFT-Calculated Structures in Solution and Single-Crystal X-ray Method

The single-crystal X-ray structures of chrysophanol (**1**) [54], emodin (**2**) [55], and physcion (**3**) [56] show that the C(1)–OH and C(8)–OH groups interact with the C(9)=O oxygen atom of the carbonyl group forming an intramolecular hydrogen bond. This is in agreement with our experimental NMR data and DFT-calculated structures. The DFT structures demonstrate excellent agreement with the crystallographic distances of the heavy atoms. Thus, for emodin the O(1)∙∙∙O(8) distances are identical to ±0.006 Å in both structural methods. On the contrary, significant differences were observed for distances involving hydrogen atoms. Thus, the X-ray hydrogen bond distances are H(1)∙∙∙O(9) = 1.831 Å and H(8)∙∙∙O(9) = 1.849 Å while in the DFT-calculated structures (with a discrete molecule of DMSO in CPCM-DMSO) the respective distances were found to be H(1)∙∙∙O(9) = 1.6887 Å and 1.6958 Å and H(8)∙∙∙O(9) = 1.6895 Å and 1.6944 Å for conformers A_2_ and B, respectively. When the X-ray structures were used as input geometries, the DFT-computed ^1^H-NMR chemical shifts of C(1)–OH and C(8)–OH groups were found to strongly deviate from the experimental values by up to 8.84 ppm (Table 4). The chemical shifts of the –CH (aromatics) and –CH_3_ protons were also found to strongly deviate up to 3.7 ppm from the experimental values.

Figure 11 illustrates excellent agreement of the DFT-calculated ^1^H-NMR chemical shifts of the energy optimized structures of chrysophanol (**1**), emodin (**2**), and physcion (**3**) with the experimental values. On the contrary, the ^1^H-NMR chemical shifts of the X-ray structures, without optimization, of chrysophanol (**1**), emodin (**2**), and physcion (**3**), strongly deviate from the experimental values. The results of the linear regression analysis (R^2^ = 0.4147, intercept = 0.4960 and slope = 0.4467) can be compared with the DFT-calculated energy-minimized structures (R^2^=0.9987 and 0.9976, intercept = 0.1686 and 0.2376, and slope = 1.0138 and 1.0026 for conformers A and B, respectively). In the above analysis, the ^1^H-NMR chemical shifts of the C(3)–OH were not taken into consideration due to significant solvent effects which were analysed in Section 2.2.3. Comparison of the X-ray structures with those obtained with DFT calculations, demonstrates significant differences for all distances involving hydrogen atoms (Appendix A). Thus, the X-ray O–H bond distances are shorter by 0.152 to 0.132 Å and the C–H bond lengths are shorter by 0.133 to 0.100 Å than those obtained with DFT calculations. The above analysis demonstrates that the X-ray structures do not provide reliable results for hydrogen bond and C–H bond lengths. It is well known that, due to high electronegativity of the oxygen atom and libration effects, the O–H bonds appear too short in X-ray structure determination [1]. This results in poor correlation between X-ray and the more accurate neutron diffraction data [57,58].

Differences between the X-ray and DFT-calculated structures have also been observed in the conformation of the substituents. Thus, a torsion angle C(5)–C(6)–C(6′)–H(6′) = −164° was observed in the X-ray structure of physcion (**3**) [56] which deviates from the DFT value of −179.75°, due to π-stacking interaction of the hydrogen atom with an aromatic system. In the X-ray structure of chrysophanol (**1**) [54] a particular conformer could not be observed due to fast rotation of the CH_3_– group. In the X-ray structure of physcion (**1**) [56] the –OCH_3_ group adopts conformation A, while in the DFT-calculated structure, conformer B is the most stable one.

## 3. Materials and Methods

### 3.1. Chemicals

Chrysophanol (**1**), emodin (**2**), and physcion (**3**) were obtained from Molecular Data Bank, ICCBS, University of Karachi. DMSO-d_6_, acetone-d_6_, and CDCl_3_ were purchased from Armar Chemicals AG, Dottingen, Switzerland and TFA from Sigma-Aldrich, Darmstadt, Germany.

### 3.2. NMR

NMR experiments were performed on Bruker AV-spectrometers (400, 500, and 800 MHz, Bruker, Billerica, MA, USA) equipped with TXI cryoprobes. Samples were dissolved in 0.6 mL of deuterated solvent and transferred to 5 mm NMR tubes. Chemical shifts were measured with reference to the residual proton signal of the incompletely deuterated solvent. 1D selective TOCSY and NOE and 2D ^1^H-^13^C HSQC and HMBC experiments were carried out using standard pulse program of Bruker.

### 3.3. Computational Methods

The computational study was performed by using the Gaussian 09 with the DFT method [52]. The structures were minimized/optimized by using five functionals and three basis sets: B3LYP/6-31 + G(d), ωB97XD/6-31 + G(d), APFD/6-31 + G(d), M06-2X/Def2TZVP, and TPSSh/TZVP. The ^1^H-NMR chemical shifts were calculated with the GIAO method by using the B3LYP/6-311 + G(2d,p) level with the CPCM (conductor like polarizable continuum model) [53]. Computations were also performed in the case of emodin (**2**) by the inclusion of discrete solvent molecules of DMSO, acetone, and CHCl_3._ The scanning of torsional and dihedral angles was performed by using the redundant coordinates in Gaussian 09. The optimized geometries were verified as minimized geometries by performing the frequency calculation at the same level (zero imaginary frequencies). TMS was used as reference for the computed ^1^H-NMR chemical shifts and was optimised at the same level.

## 4. Conclusions

From the data reported herein, it can be concluded that:
Excellent linear correlation can be obtained between experimental and DFT-calculated ^1^H-NMR chemical shifts even with computationally less demanding level of theory.Inclusion of discrete solvent molecules induces a minor effect on the computed ^1^H-NMR chemical shifts of the intramolecular hydrogen bond, but shows a significant effect on the ^1^H-NMR chemical shifts of the C(3)–OH which participates in intermolecular solute-solvent hydrogen bond; this results in excellent agreement with the experimental ^1^H-NMR chemical shifts.The ^1^H-NMR chemical shifts of the OH groups which participate in intramolecular hydrogen bond are dependent on the conformational state of substituents and, thus, can be used as molecular sensors in conformational analysis.The use of X-ray structures as input geometries results in ^1^H-NMR chemical shifts which strongly deviate from the experimental values and no functional dependence could be obtained.Comparison of the most important intramolecular data of the DFT-calculated and the X-ray structures demonstrate very good agreement with distances involving heavy atoms but significant differences for distances involving hydrogen atoms, most notably the intramolecular hydrogen bond and C–H bond lengths which deviate by 0.152 to 0.132 Å and 0.133 to 0.100 Å, respectively. Further differences have been found in the conformational state of the –CH_3_, –OCH_3_, and –OH groups.

The great sensitivity, therefore, of ^1^H-NMR chemical shifts to hydrogen bond properties, solute–solvent interactions, torsion angle, and C–H bond lengths can provide an excellent method for obtaining high-resolution structures in solution.

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
