# Peer review of "Solvent-Dependent Structures of Natural Products Based on the Combined Use of DFT Calculations and 1H-NMR Chemical Shifts"

_molecules, 2019, doi:10.3390/molecules24122290_

Round 1

Reviewer 1 Report

The manuscript “Solvent-dependent DFT-calculated structures of natural products based on 1H-NMR chemical shifts vs. single-crystal X-ray structures” main focus is the 1H NMR chemical shifts of the hydroxyl group of three molecules. The dependence of the 1H NMR chemical shifts of the OH group due to intra and intermolecular interactions are described. The manuscript is well written but the title could be improved. The X-ray had no utility in the manuscript, other than to provide a structure to calculate the 1H NMR chemical shifts, and to show that the structure provided by the X-ray is not adequate to calculate the chemical shifts. This is understandable because the X-ray is of the crystalline compound, and the NMR is of the material in solution.

What do you mean by “Solvent-dependent DFT-calculated”?

- DFT calculation depends on the solvent? DFT can not be done without solvent?

Suggestion: Please choose another title for your manuscript.

Please find below some suggestions I think could improve the manuscript.

Page 2, Scheme 1: Please replace the R group in the structures by CH3. Unless the R group is different for each structure.

Page 6, line 158: What is ppb? What is -14.0 ppb? Please explain.

Page 9, line 267: C(4)–C(3)–O(3)–C(3).

Suggestion: C(4)–C(3)–O(3)–H(3).

Page 13, lines 333-335: Chyrosophanol (1), emodin (2), and physcion (3) exist in a single conformational state, due to rotation of methyl group substituent in which one of the C–H groups is in the eclipsed conformation with respect to the planar aromatic system. The electronic energy of the transition state was found to be only 0.18 kcal/mol (ΔG = 0.85 kcal/mol) higher than that of the ground state (Fig. S4).

Suggestion: What do you mean by “Chyrosophanol (1), emodin (2), and physcion (3) exist in a single conformational state, due to rotation of methyl group substituent …”?

If the methyl group could not rotate more conformers could exist?

Why do you need to state this? “The electronic energy of the transition state was found to be only 0.18 kcal/mol (ΔG = 0.85 kcal/mol) higher than that of the ground state (Fig. S4).” The electronic energy of the transition state had any use?

Page 19, line 505: programme

Suggestion: program

Page 19, line 508: compounds

Suggestion: structures

B3LYP6-31G+d ; APDF/631G+d ; ωB97XD/6-31G+d

Suggestion: Please correct all over the manuscript and supplementary material.

Please write as follows:

B3LYP/6-31+G(d)

APDF/631+G(d)

ωB97XD/6-31+G(d)

Author Response

Reviewer #1

(1) The manuscript “Solvent-dependent DFT-calculated structures of natural products based on 1H-NMR chemical shifts vs. single-crystal X-ray structures” main focus is the 1H NMR chemical shifts of the hydroxyl group of three molecules. The dependence of the 1H NMR chemical shifts of the OH group due to intra and intermolecular interactions are described. The manuscript is well written

We are very pleased with this comment of the Reviewer.

(2)….but the title could be improved. The X-ray had no utility in the manuscript, other than to provide a structure to calculate the 1H NMR chemical shifts, and to show that the structure provided by the X-ray is not adequate to calculate the chemical shifts. This is understandable because the X-ray is of the crystalline compound, and the NMR is of the material in solution.

“What do you mean by “Solvent-dependent DFT-calculated”?

- DFT calculation depends on the solvent? DFT can not be done without solvent?”

Suggestion: Please choose another title for your manuscript.

We agree with this critical comment of the Reviewer about the ambiguity in the title of our manuscript. The title of the revised manuscript, therefore, was changed as follows: “Solvent – dependent structures of natural products based on the combined use of DFT calculations and 1H-NMR chemical shifts”. A similar correction was made also in the text (page 2, lines 66-70).

(3) “Page 2, Scheme 1: Please replace the R group in the structures by CH3. Unless the R group is different for each structure.

We agree with the Reviewer, therefore, the appropriate correction has been made in Scheme 1 of the revised manuscript.

(4) “Page 6, line 158: What is ppb? What is -14.0 ppb? Please explain.

We agree with the Reviewer, therefore, the appropriate definition was added on page 5, lines 146-147 of the revised version of our manuscript.

(5) “Page 9, line 267: C(4)–C(3)–O(3)–C(3).

We thank the Reviewer for pointed out this error in the original manuscript. The appropriate correction has been made on page 8, lines 250-251 of the revised version of our manuscript.

(6) “Page 13, lines 333-335: Chyrosophanol (1), emodin (2), and physcion (3) exist in a single conformational state, due to rotation of methyl group substituent in which one of the C–H groups is in the eclipsed conformation with respect to the planar aromatic system. The electronic energy of the transition state was found to be only 0.18 kcal/mol (ΔG = 0.85 kcal/mol) higher than that of the ground state (Fig. S4).

If the methyl group could not rotate more conformers could exist?.

Why do you need to state this? “The electronic energy of the transition state was found to be only 0.18 kcal/mol (ΔG = 0.85 kcal/mol) higher than that of the ground state (Fig. S4).” The electronic energy of the transition state had any use?.

We wish to thank the Reviewer for this critical comment. In the revised version of our manuscript (page 12, lines 317-319) the discussion of the electronic energy has been eliminated and the important ΔG term is discussed according to the recommendation of the Reviewer in terms of the minimum energy conformational state of the CH3 group rather than in terms of a single conformation.

(7) “Page 19, line 505: programme.

We agree with the Reviewer, therefore, the appropriate correction has been made in the revised version of our manuscript.

(8) “Page 19, line 508: compounds.

We agree with the Reviewer, therefore, in the revised version of our manuscript the appropriate correction has been made.

(9) “B3LYP6-31G+d ; APDF/631G+d ; ωB97XD/6-31G+d.

Suggestion: Please correct all over the manuscript and supplementary material.

Please write as follows:

B3LYP/6-31+G(d)

APDF/631+G(d)

ωB97XD/6-31+G(d)

We agree with this critical comment of the Reviewer, therefore, the appropriate corrections have been made all over the revised version of our manuscript, in Figure 3 and in the Supplementary Material.

Reviewer 2 Report

COMMENTS:

The manuscript (MS) by Mari et al. describes some 1H NMR experimental and DFT calculation results on solvent-dependent structures of three natural products (compounds 1-3 belonging to the anthraquinone family) vs. their single-crystal X-ray structures. The results are interesting and the subject of the work is important because it broadens our knowledge about intra- and intermolecular hydrogen bonds in solution and solid state, reflected by routine experimental and theoretical methods. So, this material is potentially publishable in Molecules. However, the MS in its current version contains a lot of errors and shortcomings of the general and/or detailed nature. Hence, I recommend this work for publication in Molecules after a major revision, i.e., after consideration all of the issues mentioned below.

Major Points:

1.         The MS and Supplementary Material (SM). The authors use the name chyrosophanol for compound 1 even though it is commonly known as chrysophanol or chrysophanic acid. Therefore, its name must be corrected.

2.         The whole MS contains a lot of repetitions and some data should rather be presented only in tables (e.g., rows 114-116 and 176/177).

3.         I think that almost the whole MS contains “6-31G+d” instead of the 6-31G+G(d) basis set. This mistake must be corrected.

4.         The recorded 13C NMR spectra of all compounds 1-3 should be given in SM [at least, in the two text versions (CDCl3 and DMSO-d6)].

5.         Row 152:  The standard notation of an isotope effect under consideration is nΔH(D).

6.         Row 155:  In general, the symbols 1H and 2H (and not 2D) are in common use.

7.         Rows 200-201: The authors should explain what dictated the choice of five density functionals used. Moreover, a whole series of these functionals should be applied to proper investigate the effect of the basis set on the geometry optimization and the quality of the 1H NMR chemical shift calculations. Typically, these two stages of the study are delineated - otherwise, it is very difficult to compare the results obtained (the case under consideration).

8.         Rows 205, 214, 231, 492, and 540: minimization of the structures/geometries ® optimization of the structures. To obtain an optimized geometry (structure), energy minimization is carried out.

9.         The Cartesian coordinates of “the best” calculated structures also those with discrete molecules of the solvent (all given in SM) are not absolutely required, but it would be very easy to reproduce the calculation results obtained by the authors.

Minor Points:

1.         Rows 97-102: In my opinion, an inclusion of the discussed HMBC spectrum of 1 in SM would be much more elegant.

2.         Figures 2 and 3: The used operating frequency should be given in legends to the figures.

3.         Row 162: 2D isotope effect ® 2H or D isotope effect (see #5 above).

4.         Rows 164/170: shifts, Δδ/ΔT, of OH groups can ® … shifts of OH groups, Δδ/ΔT, can

5.         Row 251:  the statement μΑ = 7μΒ should be discussed in more detail.

6.         Rows 209/508:  I think that the Def2TZVP base differs from TZVP. Therefore, the three (and not two) basis sets were employed really.

7.         Rows 474-478:  In the discussion of results a more recent paper in this field (Cryst. Growth Des. 2016, 16, 6841) should also be considered and cited.

Author Response

Reviewer #2

(1) The manuscript (MS) by Mari et al. describes some 1H NMR experimental and DFT calculation results on solvent-dependent structures of three natural products (compounds 1-3 belonging to the anthraquinone family) vs. their single-crystal X-ray structures. The results are interesting and the subject of the work is important because it broadens our knowledge about intra- and intermolecular hydrogen bonds in solution and solid state, reflected by routine experimental and theoretical methods. So, this material is potentially publishable in Molecules.

We are very pleased with this comment of the Reviewer.

(2) However, the MS in its current version contains a lot of errors and shortcomings of the general and/or detailed nature. Hence, I recommend this work for publication in Molecules after a major revision, i.e., after consideration all of the issues mentioned below.

(a) “The MS and Supplementary Material (SM). The authors use the name chyrosophanol for compound 1 even though it is commonly known as chrysophanol or chrysophanic acid. Therefore, its name must be corrected

We thank the Reviewer for this comment which is similar to the comment of the Reviewer 3, therefore, the appropriate corrections have been made in the revised version of our manuscript.

(b) “The whole MS contains a lot of repetitions and some data should rather be presented only in tables (e.g., rows 114-116 and 176/177.”

Rows 97-102: In my opinion, an inclusion of the discussed HMBC spectrum of 1 in SM would be much more elegant.”

The recorded 13C NMR spectra of all compounds 1-3 should be given in SM [at least, in the two text versions (CDCl3 and DMSO-d6)].

We agree with the Reviewer, therefore, rows 93-94, 97-99, 100-102, 103-104, 113-117, 122, 170-171, 176-180, 367-369 and Figure 1 (which is very similar to Figure 2) were eliminated. Furthermore, following the recommendation of the Reviewer, a 1H-13C HMBC spectrum (Figure S1) and 13C NMR data (Table S1) are included in the Supplementary Material.

I think that almost the whole MS contains “6-31G+d” instead of the 6-31G+G(d) basis set. This mistake must be corrected

We thank the Reviewer for this comment which is similar to comment 9 of the Reviewer 1, therefore, the appropriate corrections have been made in the revised version of our manuscript.

 (c) “Row 152:  The standard notation of an isotope effect under consideration is nΔH(D)"

We agree with the Reviewer, therefore, the appropriate correction has been made in the revised version of our manuscript.

(d) “Row 155:  In general, the symbols 1H and 2H (and not 2D) are in common use.”

We agree with the Reviewer, therefore, the appropriate corrections have been made in the revised version of our manuscript.

(e) “Rows 200-201: The authors should explain what dictated the choice of five density functionals used. Moreover, a whole series of these functionals should be applied to proper investigate the effect of the basis set on the geometry optimization and the quality of the 1H NMR chemical shift calculations. Typically, these two stages of the study are delineated - otherwise, it is very difficult to compare the results obtained (the case under consideration)

We wish to thank the Reviewer for this critical comment. As pointed out in a comprehensive review article of Tantillo et al., Chem. Rev., 2012, 112, 1839-1862 (ref [20] in the revised version of our manuscript), increasing basis set size (and computational time) does not necessarily result in more accurate chemical shifts, at least with B3LYP. Furthermore, computed geometries turn out to be relatively consistent irrespective of the computational method used. This is in excellent agreement with the results of out manuscript. Therefore, the appropriate text has been added in the revised version of out manuscript (page 6, lines 194-198).

(f) “Rows 205, 214, 231, 492, and 540: minimization of the structures/geometries ® optimization of the structures. To obtain an optimized geometry (structure), energy minimization is carried out.”

We agree with the Reviewer, therefore, the appropriate correction has been made in the revised version of our manuscript.

(g) “The Cartesian coordinates of “the best” calculated structures also those with discrete molecules of the solvent (all given in SM) are not absolutely required, but it would be very easy to reproduce the calculation results obtained by the authors.”

The Cartesian coordinates of “the best” calculated structures (total of 27 pages) are provided in the Supplementary Material of the revised version of our manuscript.

(h) “Figures 2 and 3: The used operating frequency should be given in legends to the figures"

We agree with the Reviewer for this comment, therefore, the operating frequencies of Figures 1 and 2 and Figure S1 are provided in the revised version of our manuscript.

(i) “Row 162: 2D isotope effect ® 2H or D isotope effect (see #5 above)

We agree with the Reviewer for this comment, therefore, the appropriate corrections have been made in the revised version of our manuscript.

(g) “Rows 164/170: … shifts, Δδ/ΔT, of OH groups can ® … shifts of OH groups, Δδ/ΔT, can

We wish to thank the Reviewer for this comment, therefore, in the revised version of our manuscript the repetition was eliminated (page 5, lines 156-158).

(k) “Row 251:  the statement μΑ = 7μΒ should be discussed in more detail

We agree with the Reviewer, therefore, the above statement is discussed (page 8, lines 234-235) in the revised version of our manuscript.

(l) “Rows 209/508:  I think that the Def2TZVP base differs from TZVP. Therefore, the three (and not two) basis sets were employed really

We wish to thank the Reviewer for pointed out this inconsistency, therefore, in the revised version of our manuscript we emphasize that three basis sets (not two) were used.

(m) “Rows 474-478:  In the discussion of results a more recent paper in this field (Cryst. Growth Des. 2016, 16, 6841) should also be considered and cited

We agree with the Reviewer, therefore, the suggested recent paper in this field is included in the revised version of our manuscript. (Reference 58, page 16, lines 455-458).

Reviewer 3 Report

In my opinion, the manuscript is clear and it's scientific content is almost free of any major mistakes.

I consider that the manuscript could be suitable for publication in Molecules if the authors revise carefully it in order to correct mistakes like te use of chyrosophanol instead chrysophanol.

Author Response

Reviewer #3

In my opinion, the manuscript is clear and it's scientific content is almost free of any major mistakes.

We are very pleased with this comment of the Reviewer.

I consider that the manuscript could be suitable for publication in Molecules if the authors revise carefully it in order to correct mistakes like te use of chyrosophanol instead chrysophanol

This comment is similar to that of the Reviewer 2, therefore, the appropriate correction has been made in the revised version of our manuscript.